# Neuropsychiatric safety of varenicline in the general and COPD population with and without psychiatric disorders: a retrospective cohort study in a real-world setting

Yuanyuan Wang ,[1] Jens H. Bos,[1] Catharina C.M. Schuiling-Veninga,[1] H. Marike Boezen,[2,3] Job F. M. van Boven [3,4] Bob Wilffert,[1,4] Eelko Hak[1,2]

For numbered affiliations see end of article.

**Correspondence to**
Dr Yuanyuan Wang;
yuanyuanwang.research@gmail.com

## ABSTRACT

**Objectives** To evaluate the real-world association between varenicline and neuropsychiatric adverse events (NPAEs) in general and chronic obstructive pulmonary disease (COPD) population with and without psychiatric disorders compared with nicotine replacement therapy (NRT) to strengthen the knowledge of varenicline safety.

**Design** A retrospective cohort study.

**Setting** Prescription database IADB.nl, the Netherlands.

**Participants** New users of varenicline or NRT among general (≥18 years) and COPD (≥40 years) population. Psychiatric subcohort was defined as people prescribed psychotropic medications (≥2) within 6 months before the index date.

**Outcome measures** The incidence of NPAEs including depression, anxiety and insomnia, defined by new or naive prescriptions of related medications in IADB.nl within 24 weeks after the first treatment initiation of varenicline or NRT.

**Results** For the general population in non-psychiatric cohort, the incidence of total NPAEs in varenicline (4480) and NRT (1970) groups was 10.5% and 12.6%, respectively (adjusted OR (aOR) 0.85, 95% CI 0.72 to 1.00). For the general population in psychiatric cohort, the incidence of total NPAEs was much higher, 75.3% and 78.5% for varenicline (1427) and NRT (1200) groups, respectively (aOR 0.82, 95% CI 0.68 to 0.99). For the COPD population (1598), there were no differences in the incidence of NPAEs between comparison groups in both the psychiatric cohort (aOR 0.97, 95% CI 0.66 to 1.44) and non-psychiatric cohort (aOR 0.81, 95% CI 0.54 to 1.20). Results from subgroup or sensitivity analyses also did not reveal increased risks of NPAEs but showed decreased risk of some subgroup NPAEs associated with varenicline.

**Conclusions** In contrast to the concerns of a possible increased risk of NPAEs among varenicline users, we found a relative decreased risk of total NPAEs in varenicline users of the general population in psychiatric or non-psychiatric cohorts compared with NRT and no difference for NPAEs between varenicline and NRT users in smaller population with COPD.

## Strengths and limitations of this study

► The use of large real-world population data makes the findings from this study more applicable to daily clinical practice.

► The risk of neuropsychiatric adverse events associated with varenicline were explored in both the general population and potential patients with chronic obstructive pulmonary disease (COPD).

► Subgroup and sensitivity analysis were performed to confirm the robustness and generalisability of the study findings.

► Despite the control for potential measured confounding and stratification according to risk groups, unmeasured healthy user bias may still be present but is unlikely to mask substantial risks of varenicline.

► Behavioural changes like self-harm or suicide could not be evaluated in this study.

## INTRODUCTION

Tobacco smoking is the leading preventable risk factor for a range of physical and mental illnesses,[1–4] which poses enormous threats to global public health.[5] Although average global smoking rates have declined since 1990 through tobacco control policies,[6] the actual number of smokers and disease burden related to smoking continues to increase owing to population growth.[7] More than 8 million people are killed by tobacco use each year.[8] Therefore, more intensified efforts are needed to fight this deadly epidemic. Smoking cessation strategies as key interventions to prevent smoking-related diseases are therefore urgently needed.[9] Varenicline was the first-line non-nicotine pharmacotherapy for smoking cessation and has greater efficacy than bupropion, single form nicotine replacement therapy (NRT) or placebo.[10 11] However, substantial concerns regarding its

neuropsychiatric safety (eg, suicidal thoughts and aggressive behaviour) have been raised by case reports since its approval in the USA in 2006.[12] Therefore, after the first safety communication and public health advisory in 2008, the US Food and Drug Administration (FDA) released a black box warning on 1 July 2009.[13] Of note, the early safety concerns regarding varenicline were from uncontrolled case reports, which could not establish the causality because of a lack of control or comparator. Afterwards, many randomised controlled trials (RCTs) were conducted to evaluate the possible risk of neuropsychiatric adverse events (NPAEs). Notably, pooled evidence of these RCTs did not indicate an association between varenicline and NPAEs.[14] Neither did the Evaluating Adverse Events in a Global Smoking Cessation Study show a significant increase in NPAEs with varenicline relative to NRT or placebo.[10]

Although the black box warning in varenicline's product labelling was already removed by both FDA and European Medicines Agency in 2016, there are still concerns among patients and physicians about its neuropsychiatric safety resulting from the limited external validity of existing evidence from RCTs,[15] especially for high-risk population with increased smoking prevalence, such as those with COPD and psychiatric disorders.[16] Importantly, COPD patients are older and suffer from many comorbidities making these patients more susceptible to drug–drug interactions potentially leading to related adverse drug events (ADEs).[17] Similarly, it has been reported that individuals with psychiatric disorders are prone to experience relapse of psychiatric symptoms.[18 19] Although many RCTs and post hoc analyses of some RCTs showed consistent results about varenicline safety in subjects with specific psychiatric disorder,[20–27] real-world varenicline safety among these high-risk population is only explored in few studies,[28 29] several related large observational studies of high-quality were all about general population, rather than specific high-risk population.[30–33] It is important to further investigate the psychiatric safety of varenicline in special patients with COPD or psychiatric disorder or both in real-world setting to strengthen the knowledge of varenicline safety.

We therefore conducted this cohort study based on real-life data to assess the risk of NPAEs in starters with varenicline versus NRT starters in both the general and the chronic obstructive pulmonary disease (COPD) population with and without psychiatric disorders.

## METHODS
### Study design and setting
We conducted a retrospective cohort study based on the University of Groningen pharmacy dispensing database IADB.nl (http://www.iadb.nl/), which has been widely used for various drug utilisation studies.[34] IADB.nl contains information of prescribed medications from 70 community pharmacies covering a representative population of approximately 700 000 persons of the Netherlands,

regardless of insurance type. It provides both patient information (eg, date of birth and gender) and complete prescription records including the date of dispensing, quantity dispensed, dose regimen, the number of days the drug will be used and the related Anatomical Therapeutic Chemical (ATC) code. The ATC code is a unique code assigned to a medicine according to the organ or system it works on and how it works. The ATC classification system proposed by the WHO is a widely accepted drug classification scheme.

### Study population
We included adult patients (≥18 years) who started with varenicline or NRT. The individuals only prescribed varenicline (ATC code: N07BA03) or NRT (N07BA01) were included as antismoking drug users from the general population. Individuals that were prescribed drugs for obstructive airway disease (R03) at least three times within 1 year since first prescription at or above the age of 40 years were defined as patients with COPD, patients with COPD who were prescribed varenicline or NRT were included as COPD antismoking drug users.

For both the general and COPD population with antismoking drug use, the first prescription date of varenicline and NRT was set as entry date (index date) of participants for exposure and control groups, respectively. To exclude the influence of other smoking cessation drugs on NPAEs, those who were prescribed other smoking cessation drugs including bupropion (N06AX12), nortriptyline (N06AA10) and cytisine (N07BA04) rather than the studied drugs (varenicline and NRT) within 180 days before or 180 days after index date were excluded. In order to get enough prescription records to define comorbidities and NPAEs, those who were registered in IADB.nl less than 6 months before or after index date were also excluded. For individuals who were prescribed both varenicline and NRT and met criteria of both groups (see figure 1), we allocated the study subject to the group with the first index date according to the intention-to-treat principle.

In both the general and the COPD population, we classified the individuals into a psychiatric cohort and non-psychiatric cohort according to the presence of psychiatric disorders defined by the prescription of two or more drugs of analgesics (ACT codes N02, see online supplemental table S1), antiepileptics (N03), antiparkinson drugs (N04), psycholeptics (N05) or psychoanaleptics (N06) within 6 months before the index date. In all, our study population covered four separate cohorts ((1) general population in psychiatric cohort, (2) general population in non-psychiatric cohort, (3) COPD population in psychiatric cohort and (4) COPD population in non-psychiatric cohort) in which the association between exposure and outcomes were assessed (figure 1).

### Exposure and outcomes
We defined individuals using varenicline as the exposure group and those using NRT as the reference group. The

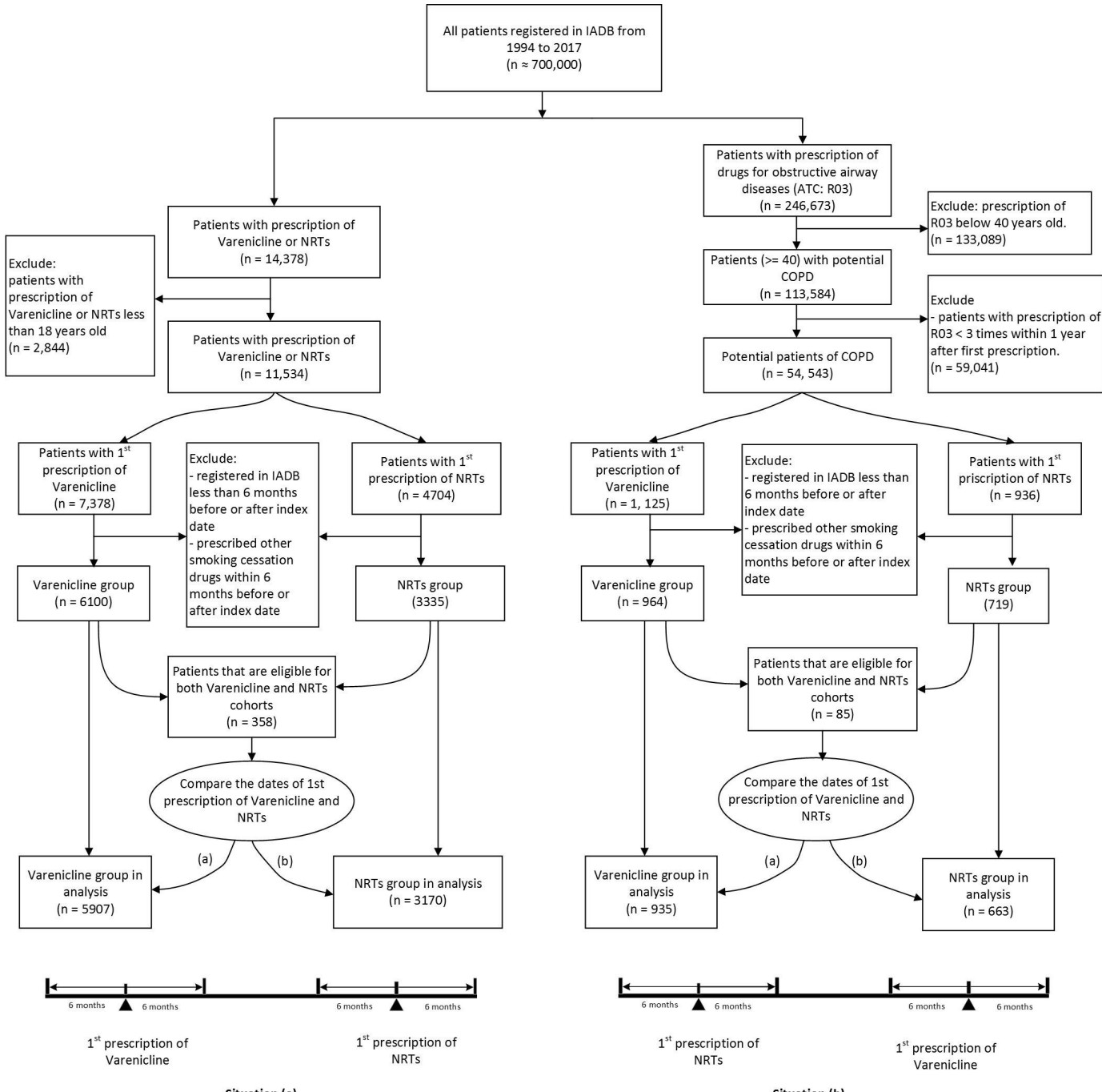

**Figure 1** Flow chart of population selection. ATC, Anatomical Therapeutic Chemical; NRT, nicotine replacement therapies.

primary outcome was incidence of any drug-treated NPAE including depression (ATC codes N06: antidepressants; N06CA: antidepressants in combination with psycholeptics), anxiety (N05B: anxiolytics) and insomnia (N05C: hypnotics and sedatives; see online supplemental table S1), defined as one or more new or naive prescriptions of the specified drugs mentioned previously within 24 weeks after index date. Considering the use of varenicline could be possibly extended to 24 weeks, which is also a suggested observed period by previous related studies.[28] Those who used the drugs under previously mentioned ATC codes level for NPAEs that prescribed before index

date but without new prescription after index date were not counted as people with NPAE. Those who never be prescribed drugs under previously mentioned ATC codes level before index date but were prescribed (naive prescription) after index date will be included as people with NPAE.

### Covariates
The following covariates were included as possible confounders: age, gender, social economic status (SES) based on postal codes, previous psychiatric disorders and other comorbidities including heart failure,

ischaemic heart disease, hypertension, cancer, diabetes, osteoporosis, peptic ulcer and gastro-oesophageal reflux disease, rheumatic arthritis, thyroid disorder, anaemia, glaucoma, gout and allergic rhinitis. All the comorbidities were defined by at least two prescriptions of related drugs within 6 months prior to index date (see detailed ATC codes in online supplemental table S1), The ATC codes used to define the comorbidities in this study were consistent with those in previous published papers.[35–39]

### Statistical methods

The continuous and categorical variables are presented as means with SD and numbers with percentages, respectively. The differences of characteristics between two groups were compared using Student's t-tests and $\chi^2$ tests for continuous and categorical variables, respectively. Binary logistic regression modelling was used to obtain the OR and corresponding 95% CI after adjustment for the potential confounders. A two-sided p≤0.05 was considered to be statistically significant for all tests. All analyses were performed using IBM SPSS Statistics V.25 (IBM Corporation, Armonk, New York, USA) for Windows.

### Subgroup and sensitivity analyses

Considering the possible influence of age and gender on the association between NPAEs and varenicline use, we further analysed the primary outcome in the four cohorts (general and COPD populations with or without psychiatric disease, separately) by stratifying the results by age groups and gender. To further test the robustness of the results, we performed several sensitivity analyses. First, because the possible treatment duration of varenicline is just 12 weeks, we explored the outcome occurring within 12 weeks after treatment initiation. Second, to exclude an active psychiatric status with possible frequently medication change, we further limit the study population only to those subjects who were not prescribed any drugs for psychiatric disorders within 30 days before index date. Third, considering the influence of policy changes about reimbursement of smoking cessation treatment on the use of antismoking drugs,[40] we also performed a sensitivity analysis by excluding the patients whose prescription date of varenicline or NRT may be in the period of Dutch smoking policy changes (ie, from 1 July 2011 to 30 June 2013). Finally, to exclude the possible selection bias for treatments, we also apply the inverse probability of treatment weighted (IPTW) regression to further test the robustness of our main results.

### Patient and public involvement

Patients were not involved in the design or conduction of the study, as this research was based on established cohort and previously collected data. The results of this study will be distributed in scientific reports or academic conferences to benefit policy makers, clinicians and patients.

## RESULTS

### Baseline characteristics

In total, we included 9077 subjects who initiated varenicline or NRT from the general population, of which 2627 had psychiatric disorders (varenicline vs NRT: 1427 vs 1200), while 6450 did not have psychiatric disorder (varenicline vs NRT: 4480 vs 1970). For the COPD population, we included 1598 individuals, of which 649 had psychiatric disorders (varenicline vs NRT: 327 vs 322) and 949 did not have psychiatric disorders (varenicline vs NRT: 608 vs 341). In both the general and COPD population, individuals treated with varenicline were younger than those treated with NRT (table 1). Drug use for heart failure and ischaemic heart disease was lower in the varenicline-treated than the NRT-treated group. In patients without psychiatric disorders, drug use for other comorbidities (eg, diabetes and osteoporosis) was also less in individuals treated with varenicline than in those with NRT.

### Primary outcome in the general and COPD population

In the general population with psychiatric disorders, the incidence of overall NPAEs within 24 weeks was lower in the varenicline group than the NRT group (75.3% vs 78.5%, OR 0.83 (95% CI 0.69 to 1.00), table 2). After adjusting for potential confounders, the association did not substantially change (adjusted OR (aOR) 0.82, 95% CI 0.68 to 0.99). All the specific NPAEs were also less in the varenicline group than the NRT group, although the difference between the two comparison groups for depression and insomnia events did not reach statistical significance.

In the general population without psychiatric disorders, the incidence rates of NPAEs were lower than among those with psychiatric disorders. The incidence of overall NPAEs within 24 weeks in the varenicline group was lower than those in the NRT group (10.5% and 12.6%, respectively; aOR 0.85, 95% CI 0.72 to 1.00)). No difference was observed between the two treatment groups for depression and anxiety; however, less insomnia was seen for varenicline than NRT (aOR 0.63, 95% CI 0.49 to 0.82).

In the COPD population, we did not see a statistically significant difference for incidence of overall NPAEs between the varenicline and NRT groups for both the psychiatric cohort (aOR 0.97, 95% CI 0.66 to 1.44, table 3) and the non-psychiatric cohort (aOR 0.81, 95% CI 0.54 to 1.20). There were also no differences for specific NPAEs between treatment groups in these two cohorts except for anxiety, which was observed significantly less in the varenicline group compared with the NRT group (aOR 0.68, 95% CI 0.49 0.94) for the psychiatric cohort only.

In the model for the primary outcome of NPAEs among overall general population, there were no significant interaction between treatments (varenicline vs NRTs) and cohorts (psychiatric vs non-psychiatric) interaction, which is also the same for the model of NPAEs among overall COPD patients.

**Table 1** Baseline characteristics of general and COPD population with and without psychiatric disorders by treatment groups

| Characteristics | General population (n=9077) | | | | COPD population (1598) | | | |
| --- | --- | --- | --- | --- | --- | --- | --- | --- |
| | Psychiatric cohort (n=2627) | | Non-psychiatric cohort (n=6450) | | Psychiatric cohort (n=649) | | Non-psychiatric cohort (n=949) | |
| | Varenicline (n=1427) | NRT (n=1200) | Varenicline (n=4480) | NRT (n=1970) | Varenicline (n=327) | NRT (n=322) | Varenicline (n=608) | NRT (n=341) |
| **Age (years)** | | | | | | | | |
| Mean (SD) | 54.5 (9.6)* | 55.8 (11.8) | 52.1 (9.7)* | 53.4 (12.5) | 59.2 (8.6)* | 62.4 (10.2) | 58.9 (8.9)* | 62.7 (9.8) |
| Age range | 32–87 | 25–92 | 32–102 | 22–90 | 41–87 | 41–92 | 40–102 | 42–89 |
| **Gender (n, %)** | | | | | | | | |
| Men | 541 (37.9)* | 523 (43.6) | 2406 (53.7) | 1100 (55.8) | 111 (33.9) | 126 (39.1) | 310 (51.0) | 184 (54.0) |
| Female | 886 (62.1) | 677 (56.4) | 2074 (46.3) | 870 (44.2) | 216 (66.1) | 196 (60.9) | 298 (49.0) | 157 (46.0) |
| **Year of index date (n, %)** | | | | | | | | |
| 1994–2010 | 519 (36.4)* | 492 (41.0) | 1497 (33.4)* | 895 (45.4) | 135 (41.3) | 135 (41.9) | 211 (34.7) | 124 (36.4) |
| 2011–2017 | 908 (63.6) | 708 (59.0) | 2983 (66.6) | 1075 (54.6) | 192 (58.7) | 187 (58.1) | 397 (65.3) | 217 (63.6) |
| **Social economic status (n, %)** | | | | | | | | |
| Low | 714 (50.0)* | 655 (54.6) | 2117 (47.3)* | 1080 (54.8) | 170 (52.0) | 172 (53.4) | 287 (47.2) | 175 (51.3) |
| High | 713 (50.0) | 545 (45.4) | 2363 (52.7) | 890 (45.2) | 157 (48.0) | 150 (46.6) | 321 (52.8) | 166 (48.7) |
| **Comorbidities (n, %)** | | | | | | | | |
| Heart failure | 41 (2.9)* | 82 (6.8) | 47 (1.0)* | 53 (2.7) | 23 (7.0)* | 42 (13.0) | 20 (3.3)* | 27 (7.9) |
| Ischaemic heart disease | 25 (1.8)* | 39 (3.3) | 36 (0.8)* | 31 (1.6) | 9 (2.8)* | 20 (6.2) | 9 (1.5)* | 12 (3.5) |
| Hypertension | 508 (35.6) | 465 (38.8) | 1039 (23.2)* | 547 (27.8) | 146 (44.6) | 167 (51.9) | 234 (38.5) | 145 (42.5) |
| Cancers | 9 (0.6)* | 0 (0.0) | 9 (0.2) | 9 (0.5) | 0 (0.0) | 0 (0.0) | 0 (0.0) | 2 (0.6) |
| Diabetes mellitus | 162 (11.4) | 129 (10.8) | 260 (5.8)* | 146 (7.4) | 53 (16.2) | 40 (12.4) | 47 (7.7)* | 43 (12.6) |
| Osteoporosis | 39 (2.7) | 32 (2.7) | 25 (0.6)* | 26 (1.3) | 17 (5.2) | 20 (6.2) | 6 (1.0)* | 14 (4.1) |
| Peptic ulcer and GORD | 436 (30.6) | 407 (33.9) | 474 (10.6)* | 253 (12.8) | 134 (41.0) | 142 (44.1) | 123 (20.2) | 82 (24.0) |
| Rheumatic arthritis | 170 (11.9) | 172 (14.3) | 204 (4.6)* | 123 (6.2) | 45 (13.8) | 41 (12.7) | 40 (6.6) | 21 (6.2) |
| Thyroid disorder | 55 (3.9) | 40 (3.3) | 96 (2.1) | 49 (2.5) | 18 (5.5) | 13 (4.0) | 18 (3.0) | 5 (1.5) |
| Anaemia | 49 (3.4) | 52 (4.3) | 49 (1.1)* | 38 (1.) | 11 (3.4) | 20 (6.2) | 4 (0.7)* | 12 (3.5) |
| Glaucoma | 13 (0.9) | 12 (1.0) | 17 (0.4)* | 23 (1.2) | 4 (1.2) | 4 (1.2) | 3 (0.5)* | 7 (2.1) |
| Gout | 4 (0.3) | 9 (0.8) | 16 (0.4)* | 24 (1.2) | 1 (0.3) | 3 (0.9) | 3 (0.5)* | 10 (2.9) |
| Allergic rhinitis | 76 (5.3) | 47 (3.9) | 104 (2.3) | 34 (1.7) | 28 (8.6) | 24 (7.5) | 33 (5.4) | 15 (4.4) |

*P<0.05.

COPD, chronic obstructive pulmonary disease; GORD, gastro-oesophageal reflux disease; NRT, nicotine replacement therapy;;

**Table 2** Incidence of neuropsychiatric adverse events (NPAEs) and association with varenicline compared with NRT in general population with and without psychiatric disorders within follow-up of 24 weeks

| NPAEs | Psychiatric cohort Varenicline (n=1427) versus NRT (n=1200) | | | Non-psychiatric cohort Varenicline (n=4480) versus NRT (n=1970) | | |
|---|---|---|---|---|---|---|
| | Events (n, %)* | Crude OR (95% CI) | Adjusted OR† (95% CI) | Events (n, %)* | Crude OR (95% CI) | Adjusted OR‡ (95% CI) |
| Overall | 1074 (75.3): 942 (78.5) | 0.83(0.69 to 1.00) | 0.82(0.68 to 0.99) | 469 (10.5): 248 (12.6) | 0.81(0.69 to 0.96) | 0.85(0.72 to 1.00) |
| Depression | 629 (44.1): 548 (45.7) | 0.94(0.80 to 1.09) | 0.90(0.77 to 1.05) | 148 (3.3): 57 (2.9) | 1.15(0.84 to 1.56) | 1.15(0.84 to 1.57) |
| Anxiety | 456 (32.0): 472 (39.3) | 0.72(0.62 to 0.85) | 0.71(0.61 to 0.84) | 215 (4.8): 110 (5.6) | 0.85(0.67 to 1.08) | 0.90(0.71 to 1.15) |
| Insomnia | 372 (26.1): 352 (29.3) | 0.85(0.72 to 1.01) | 0.89(0.75 to 1.06) | 147 (3.3): 105 (5.3) | 0.60(0.47 to 0.78) | 0.63(0.49 to 0.82) |

*Varenicline: NRT.
†Adjusted for age, gender, socioeconomic status, heart failure, ischaemic heart disease and cancer.
‡Adjusted for age, socioeconomic status, heart failure, ischaemic heart disease, hypertension, diabetes, osteoporosis, peptic ulcer and GORD, rheumatic arthritis, anaemia, glaucoma and gout.
GORD, gastro-oesophageal reflux disease; NPAEs, neuropsychiatric adverse events, defined by new or naive prescription of related medications for depression, anxiety, or insomnia within 24 weeks after index date; NRT, nicotine replacement therapy.

## Subgroup analysis

In the general population with psychiatric disorders, the risk of overall NPAEs was even lower among varenicline than NRT users for younger patients (aOR 0.54, 95% CI 0.28 to 1.04) for age <40 years, aOR 0.78, 95% CI 0.63 to 0.98 for age 40–65 years (online supplemental table S2) and females (aOR 0.74, 95% CI 0.57 to 0.97). However, in the general population without psychiatric disorders, a lower risk of overall NPAE by varenicline treatment compared with NRT was seen in older patients (aOR 0.53, 95% CI 0.34 to 0.83) and male subjects (aOR 0.78, 95% CI 0.61 to 0.99). In the COPD population, in both the psychiatric and non-psychiatric cohorts, there was no difference for overall NPAEs between the varenicline and

NRT groups in each age group and gender group (online supplemental table S2).

## Sensitivity analysis

In the general population, considering a follow-up of 12 weeks, the risk of overall NPAEs was less for varenicline compared with NRT in both the psychiatric and non-psychiatric cohorts (aOR 0.78, 95% CI 0.64 to 0.94; aOR 0.74, 95% CI 0.62 to 0.89; respectively) (online supplemental table S3). After limiting the study population to those who were not prescribed drugs for any psychiatric disorder or those who were not prescribed drugs for depression, anxiety and insomnia within 1 month before index date, there was no statistically significant difference

**Table 3** Incidence of neuropsychiatric adverse events (NPAEs) and association with varenicline compared with NRT in COPD population with and without psychiatric disorders within follow-up of 24 weeks

| NPAEs | Psychiatric cohort Varenicline (n=327) versus NRT (n=322) | | | Non-psychiatric cohort Varenicline (n=608) versus NRT (n=341) | | |
|---|---|---|---|---|---|---|
| | Events (n, %)* | Crude OR (95% CI) | Adjusted OR† (95% CI) | Events (n, %)* | Crude OR (95% CI) | Adjusted OR‡ (95% CI) |
| Overall | 258 (78.9): 260 (80.7) | 0.89 (0.61 to 1.31) | 0.97 (0.66 to 1.44) | 76 (12.5): 53 (15.5) | 0.78 (0.53 to 1.13) | 0.81 (0.54 to1.20) |
| Depression | 146 (44.6): 145 (45.0) | 0.99 (0.72 to 1.34) | 0.90 (0.65 to 1.24) | 22 (3.6): 10 (2.9) | 1.24 (0.58 to 2.66) | 1.37 (0.62 to 3.03) |
| Anxiety | 110 (33.6): 136 (42.2) | 0.69 (0.50 to 0.95) | 0.68 (0.49 to 0.94) | 38 (6.3): 25 (7.3) | 0.84 (0.50 to 1.42) | 0.86 (0.50 to 1.49) |
| Insomnia | 102 (31.2): 124 (38.5) | 0.72 (0.52 to 1.00) | 0.84 (0.60 to 1.18) | 25 (4.1): 21 (6.2) | 0.65 (0.36 to 1.19) | 0.63 (0.34 to 1.17) |

*Varenicline: NRT.
†Adjusted for age, heart failure and ischaemic heart disease.
‡Adjusted for age, heart failure, ischaemic heart disease, diabetes, osteoporosis, anaemia, glaucoma and gout.
COPD, chronic obstructive pulmonary disease; NPAEs, neuropsychiatric adverse events, defined by new or naive prescription of related medications for depression, anxiety or insomnia within 24 weeks after index date; NRT, nicotine replacement therapy.

for overall NPAEs, irrespective of presence of psychiatric disorders in the previous year (online supplemental table S4). The result was similar when we limited our cohort to individuals whose study period was not in the period of Dutch smoking policy change; the aOR for varenicline compared with NRT was 0.86, 95% CI 0.69 to 1.07 in the psychiatric cohort and 0.86, 95% CI 0.71 to 1.03 in non-psychiatric cohort.

In the COPD population, there were no statistically significant differences for overall and subgroup NPAEs within follow-up of 12 weeks between the two treatments except for anxiety in the psychiatric cohort (aOR 0.64, 95% CI 0.46 to 0.90) (online supplemental table S5). After limiting the study population to those who were not prescribed drugs for any psychiatric disorder or those who were not prescribed drugs for depression, anxiety and insomnia within 1 month before index date, there was no statistically significant difference for overall NPAEs, irrespective of presence of psychiatric disorders (online supplemental table S6). Similar results were seen when limiting individuals to those whose study period was not in the period of Dutch smoking policy change (OR 0.97, 95% CI 0.62 to 1.51; OR 1.00, 95% CI 0.64 to 1.58, respectively, for the psychiatric and non-psychiatric cohorts (online supplemental table S6).

After adjusting the results by IPTW regression in cohorts of general populations, the results (online supplemental table S4) are consistent with the results presented previously. Also, no significant change (online supplemental table S6) was observed by IPTW regression for cohorts of patients with COPD compared with results previously.

## DISCUSSION
### Main findings and interpretation
Within 24 weeks following initiation of varenicline treatment, we found no significantly increased risk of NPAEs in both the general and COPD population compared with those using NRT, irrespective of the presence of psychiatric disorders. These findings are consistent with the results of previous RCTs and large observational studies.[10 41 42] Considering the fact that the smoking cessation treatment may last for only 12 weeks without further treatment,[43] we also explored the NPAEs in this shorter time period and observed no increased risk in overall and specific NPAEs for varenicline compared with NRT.

In contrast to the concerns about a possible increased risk of NPAEs among varenicline users, we found an 18% and 15% relative decrease in NPAEs in varenicline users of the general population with and without psychiatric disorders, respectively, compared with NRT. Regarding the safety of varenicline for specific NPAEs, we recorded a 29% reduced risk of anxiety by varenicline (vs NRT) in the psychiatric cohort, and a 37% reduced risk of insomnia in the non-psychiatric cohort. Rates of depression events were comparable between the two groups in both psychiatric and non-psychiatric cohorts among the general population. These results were consistent

with the pooled results of 39 RCTs in a meta-analysis,[14] which indicated that less anxiety (HR 0.75, 95% CI 0.61 to 0.93) was also observed in the varenicline group (vs NRT), and depression episodes were also evenly distributed among two treatments (HR 0.96, 95% CI 0.75 to 1.22). Compared with our study, the difference is that in this review an increased risk of insomnia was observed in the varenicline group (HR 1.56, 95% CI 1.36 to 1.78).[14] Of note, the aforementioned review did not explore the risk of varenicline on NPAEs separately in those with and without psychiatric disorders, which may contribute to the observed differences in this review compared with our results. Of note, our result is consistent with another cohort study based on data from the general practice that a declined depression were observed to be associated with varenicline (HR 0.88, 95% CI 0.77 to 1.00)[30]; similar finding was also reported in a clinical trial.[44]

Patients with COPD are considered a high-risk population with high prevalence of smoking and relatively older age, making these persons more susceptible for possible ADEs.[17 45] Of note, in our study, we did not observe an increased risk of overall NPAEs among patients with COPD using varenicline in both the psychiatric and non-psychiatric cohorts. Of note, regarding the occurrence of specific NPAEs, less anxiety was seen in the varenicline group than in the NRT group in both cohorts. The safety of varenicline was not fully explored among patients with COPD in previous studies. To the best of our knowledge, only two studies (one RCT and one cohort study) were previously conducted.[28 46] Similar to our results, both of these two studies did not find an increased risk of NPAEs for varenicline. Notably, in the cohort study, even a reduced risk of depression was observed in varenicline users compared with users of NRT among patients with COPD.[28] This may be misled by unmeasured confounders; however, after modelling the effects of possible unmeasured confounders, the author concluded that an increased risk of these adverse events was very unlikely.[28]

It is notable that there is a large heterogeneity in the definition of NPAEs across studies. In some studies, the investigators only focused on moderate to serious adverse events like depression, suicide or mental disorders that require hospitalisation or an emergency department visit.[28 32 42 47] However, there are other studies that used all relevant adverse symptoms (eg, angry and nervousness) or adverse events such as traffic offences to define NPAEs.[10 43] In this study, we used prescriptions to define neuropsychiatric outcomes for the most commonly reported NPAEs including depression, anxiety and insomnia during the study period. Despite differences in clinical definitions, the observed 24-week event rates of specific NPAEs (3%–9%) for depression, anxiety or insomnia in our general population without psychiatric disorders were similar to previous studies.[14] However, the rate of NPAEs was substantially higher among participants in psychiatric cohort than those in non-psychiatric cohort, which was also consistent with findings from previous studies.[10 20 21 48] When we further limited our study population to those

who did not experience any psychiatric disorder or not experienced any depression, anxiety and insomnia within 1 month of enrolment, we found that both the overall NPAEs and specific NPAE reduced substantially.

Although the rates of NPAEs are different between psychiatric and non-psychiatric cohorts in this study, the presence of psychiatric disorders did not influence the risk of NPAEs by varenicline compared with NRT, which was also consistent with previous studies.[10 49] In a prospective longitudinal study among psychiatric patients, there was no exacerbation of psychiatric symptoms detected except gastrointestinal adverse events.[50] However, although it is not within the scope of this study, what need to be mentioned is that an increased rate of outpatient visits for schizophrenia was previously reported to be present only in patients with a pre-existing mental health disorder.[32] This may be explained by mediation through individual genetic liability.[51]

In interpreting the absence of increased risks by varenicline (vs NRT) observed in the general population, we can only speculate that the positive effect from varenicline may result from its effect of consistently reducing withdrawal-related symptoms of negative affect and raised levels of positive affect.[52] It has been reported that varenicline yields higher abstinence rates than NRT,[10 11] irrespective of smoker characteristics.[53] The successful quitting of smoking by varenicline may offer more benefits to the psychiatric status compared with NRT. Moreover, there is evidence that quitting smoking is associated with recovery in stress, anxiety and depression in smokers.[44 54 55] A significant and progressive improvement of anxiety and depression was also reported in an observational study, and the protective effect was observed regardless of the presence of psychiatric pathology.[49]

Many studies demonstrated gender differences in varenicline efficacy for smoking cessation[56 57]; some studies also found a difference in neuropsychiatric events between genders.[58] In this study, we found less risk of NPAEs by varenicline for females in the psychiatric cohort compared with NRT. This may be explained by the better therapeutic response to varenicline in women compared with men.[56] However, in the non-psychiatric cohort, we observed less risk of NPAEs in males by varenicline (vs NRT), which we cannot explain and more research needs to be done on gender disparities. Similarly, there are some indications for age group-dependent NPAEs risk by varenicline (vs NRT) in two cohorts of this study. In the psychiatric cohort, there were lower event rates in the varenicline group than the NRT group in younger age groups. Contrary, in the non-psychiatric cohort, lower event rates were seen in older users of varenicline than NRT. The age-specific effectiveness of varenicline relative to NRT patch or gum were also reported that only younger smokers achieved greater likelihood of abstinence than NRTs.[59] As such, age disparities also need to be studied more closely.

## Strengths and limitations

A major strength of this study is that we evaluated the safety of varenicline in both the general and COPD population based on large real-life population data making the results representative and more applicable to daily clinical practice. Both short-term (12 weeks) and long-term (24 weeks) NPAEs after treatment initiation were explored in this study. Besides the influence of current or previous psychiatric disorders, we also evaluated the influence of age and gender on the NPAEs between treatment groups. We excluded the influence of a psychiatric disorder status on the outcomes in the two comparison groups by classifying the population into two cohorts (psychiatric cohort and non-psychiatric cohort) in the stage of study design. Additionally, to test the robustness of our study results, sensitivity analyses and propensity score analysis were conducted.

Like the limitations mentioned in two previous related observational studies,[41 42] our study also has the issue of potential of unmeasured healthy user bias, although we tried our best to exclude the influence of measured baseline differences between exposure groups by adjusting for them. These unmeasured confounders could be controlled by some novel methods such as instrumental variable analysis; however, this analysis is not available in this study due to lack of a proper instrumental variable. Second, as outcome events and comorbidities were defined by prescriptions of related drugs as proxies, which is a crude measure and may have led to some misclassification. Besides, the prescription of medication may not always lead to actually intake of the drugs (non-adherence) and such misclassification, if random, may have caused associations to be biased towards the null value. Third, some serious behavioural changes like self-harm or suicide and minor symptoms could not be evaluated in this study due to the limitation of the prescription database, thus we only included the most frequently reported NPAEs that are commonly drug treated. Lastly, SES was determined by the postal code level, which could lead to some misclassification, although postal code was a frequently used predictor of SES.[60]

## CONCLUSION

From this population-based real-life cohort study, we conclude that varenicline is unlikely associated with an increased risk of NPAEs in both general and COPD patients with or without psychiatric disorders following its initiation compared with NRT. Of note, varenicline were even associated with lower risk of some subgroup NPAEs than NRTs in subcohorts of our population. Our findings are notably and consistent with previous literature. These results provide further support for the safety of varenicline to quit smoking in both the general and COPD populations.

**Author affiliations**
[1]Department of PharmacoTherapy, -Epidemiology & -Economics, Groningen Research Institute of Pharmacy, University of Groningen, Groningen, The Netherlands
[2]Department of Epidemiology, University Medical Center Groningen, University of Groningen, Groningen, The Netherlands
[3]Groningen Research Institute for Asthma and COPD (GRIAC), University Medical Center Groningen, University of Groningen, Groningen, The Netherlands
[4]Department of Clinical Pharmacy & Pharmacology, University Medical Center Groningen, University of Groningen, Groningen, The Netherlands

**Acknowledgements** The authors would like to thank IADB.nl pharmacies for generous contributions of providing their data for research.

**Contributors** YW and EH are involved in designing the study; JHB contributed the data collection, YW analysed the data and drafted the article; CCM-SV, HMB, JFMvB, BW and EH contributed to the interpretation of results and revision of the manuscript. All authors read and approved the final manuscript.

**Funding** This study was supported by internal funding. YW got the funding from China Scholarship Council (file no: 201506010259) for her PhD study.

**Competing interests** None declared.

**Patient consent for publication** Not required.

**Ethics approval** This study is based on established database IADB.nl. Data are collected in accordance with the national and European guidelines on privacy requirements for handling human data. The authors have no ethical conflicts to disclose. Ethics approval is not needed and required for this study.

**Provenance and peer review** Not commissioned; externally peer reviewed.

**Data availability statement** Data are available on reasonable request. The data analysed in this study were obtained from the IADB.nl database. Researchers interested in these data could send email to info@iadb.nl.

**ORCID iDs**
Yuanyuan Wang http://orcid.org/0000-0002-5066-1654
Job F. M. van Boven http://orcid.org/0000-0003-2368-2262

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
