## [Reviewer comments · BMJ Open]

ARTICLE DETAILS

TITLE (PROVISIONAL)	Neuropsychiatric safety of varenicline in the general and COPD population with and without psychiatric disorders: a retrospective cohort study in a real-world setting
AUTHORS	Wang, Yuanyuan; Bos, Jens; Schuilting-Veninga, Catharina; Boezen, H. Marike; van Boven, Job F. M.; Wilffert, B; Hak, Eelko

VERSION 1 – REVIEW

REVIEWER	A. Eden Evins Massachusetts General Hospital and Harvard Medical School Boston Massachusetts USA
REVIEW RETURNED	31-Jul-2020

GENERAL COMMENTS	This is a report of an apparently well-executed investigation of subsequent psychiatric medication use within 12 and 24 weeks of initiating varenicline or NRT for smoking cessation, looking particularly at smokers with COPD. It is generally clearly written. I have a number of small comments below. The primary comment I have is to improve the precision with which the outcome variable is described. The outcome is a new psychotropic medication, and should be described as such throughout. A table is needed to describe the actual outcomes (new medications) and if further inference re NPSAE is made how each new medication was interpreted (what AE each medication is inferred to be associated with). Abstract Objectives: Data on neuropsychiatric safety of varenicline is remarkably consistent outside of single case reports, particularly for high risk populations. The first sentence should be deleted. If there is a reason to believe NPSAEs would be different in those with and without COPD, by psychiatric diagnosis, it should be clarified here. Design: Please identify the comparison groups here, defining the psychiatric subcohort as people in the dataset on two or more psychotropic medications. Participants: Please clarify here: Are participants all adults in the dataset who are new users of varenicline or NRT with and without COPD? Outcome measures: please state the source of the NPSAE data here and begin to define it as a new psychotropic medication. Results: That varenicline was associated with significantly lower risk for a new psychiatric medication than the referent NRT group in several general population subsamples should be noted. Conclusion: A stronger statement such as this one from the conclusion would convey the findings better: In contrast to the concerns about a possible increased risk of NPAEs among
--

varenicline users, we found a 19% and 22% relative decrease in NPAEs in varenicline users of the general population with and without psychiatric disorders, respectively, compared with NRT and no difference in the smaller sample with COPD.

Introduction

The review of the literature has errors and should be reviewed. I believe varenicline was not the first non-nicotine pharmacotherapy for smoking cessation. The early safety concerns regarding varenicline were from uncontrolled case reports and should be so stated.

Re: related evidence from realworld setting is still lacking.

The several high quality large observational trials comparing NPSAEs in varenicline and NRT treated smokers, all of which reported finding no difference, should be cited, eg Gunnell 2009, Meyer et al 2012, the US VA and DOD observational studies.

Re: Only few studies assessed the neuropsychiatric safety of varenicline in patients with COPD or psychiatric disorders,10,20-22 and Re: results were inconsistent

Analyses of EAGLES trial NPSAEs within psychiatric subgroups vs non psychiatric controls, eg. Heffner et al 2019 and Evins et al 2020 should also be cited. These report clinically ascertained NPSAEs at a rate much lower than reported in this paper. The Chengappa et al 2014 J Clin Psych paper of a varenicline trial in 60 people with bipolar disorder should also be cited. Trials of varenicline in schizophrenia and or bipolar disorder should also be cited: Evins et al JAMA 2014, Williams et al J Clin Psych, Pachas et al 2012 J Dual Diag, and in MDD: Antheneli Ann Int Med 2013.

The NPSAE safety results from the above trials and large observational studies with NRT reference groups are remarkably consistent. The overwhelming bulk of controlled trial and observational data shows NPSAE risk of varenicline to be no different from or less than NRT and/or placebo. This is reviewed in Cather et al 2017 CNS Drugs for those with psychiatric illness and should be so stated. This does not diminish the importance of an investigation of psychiatric safety of varenicline in a COPD sample with and without evidence for psychiatric comorbidity.

Please also mention the EMA removal of the boxed warning in addition to the FDA action.

Re: we classified the individuals into a psychiatric cohort and non-psychiatric cohort according to the presence of psychiatric disorders defined by the prescription of two or more drugs from the neurological ATC group, i.e. N02, N03, N04, N05, N06 within 6 months before index date

Please list these medications.

A new psychiatric medication within 6 months of starting NRT or varenicline among those already on 2 or more psychiatric medications is a crude measure for a neuropsychiatric adverse event. This should be discussed in the limitations section.

Psychiatric medication changes are common and may be due to many reasons in addition to worsening of psychiatric symptoms, for example common reasons for changing psychiatric medications are sexual or anticholinergic side effects, sedation, extrapyramidal symptoms etc. Please refer to this outcome in the paper not as a

	NPSAE but as a new psychiatric medication. Thus depression and insomnia events would be new antidepressant medication or sedative-hypnotic medication. Also, it is not clear why pain medications N02 are included, N03 anti epileptic drugs would indicate the need for a mood stabilizer and would be associated more with bipolar disorder than depression, anxiety, or insomnia. N05 includes antipsychotics and would be associated more with bipolar disorder or schizophrenia than depression, anxiety, or insomnia. In defining the sample, those on two or more psychiatric medications are not known to have a psychiatric diagnosis. They can be referred to as those on psychiatric treatment. The difficulty here is that for example SSRIs are listed by the WHO as a treatment for depression but are often used to treat anxiety. Trazodone is an antidepressant may be used more commonly for insomnia than depression. Thus for defining the sample and the outcome, using the medications rather than the illness would improve the precision of the report. Results Those on psychiatric medication may be less likely to receive varenicline than those not on psychiatric treatment. Please report whether there is an interaction between psychiatric medication vs not and prescription of varenicline vs NRT. Discussion Of note, our result is consistent with another cohort study based data from the general practice that no increased depression were observed to be associated with varenicline (HR 0.88 [0.77-1.00]). 33 The findings are consistent also with those of Cather et al J Dual Diag 2017, which should be cited, finding varenicline treatment associated with reduced depressive symptoms. Because of higher rates of abstinence, this is also consistent with a statement later in the manuscript: Moreover, there is evidence that quitting smoking is associated with recovery in stress, anxiety and depression in smokers.^{45,46} Re: However, the rate of NPAEs was substantially higher in participants with psychiatric disorders than those without such illness, which was also consistent with findings from previous studies.³⁷⁻³⁹ This is not consistent with trials of varenicline in people with actual diagnosed psychiatric illness as cited above in a dozen studies in trials and observational data in the US and UK. Likely this is related to the definition of a medication change as a 'NPSAE' which could both be unrelated to varenicline or NRT and could often miss NPSAEs related to varenicline or NRT. Conclusion: It should be noted that not only did varenicline cause no greater new psychotropic medication use in your COPD sample of interest, it was associated with statistically significantly lower incidence of new psychotropic drug use in several subgroups. In particular, in the non-COPD psychiatric cohort, varenicline was associated with significantly fewer new psych meds than NRT. This is notable and consistent with the prior literature.
--	---

	Table 2: please list categories and individual drugs instead of disorders. Many medications can be used to treat two or more of these disorders. Serotonin and serotonin norepinephrine reuptake inhibitors may be used to treat anxiety and or depression. Sedative hypnotics may be used to treat anxiety or insomnia. Was there no new antipsychotic or mood stabilizing treatment noted?
--	--

REVIEWER	Robert Schnoll University of Pennsylvania, USA None. But I oppose naming reviewers and would not review again (I didn't realize this until I did the review).
REVIEW RETURNED	12-Aug-2020

GENERAL COMMENTS	MS Title: Neuropsychiatric safety of varenicline in the general and COPD population with and without psychiatric disorders: a retrospective inception cohort study in a real-world setting MS#: bmjopen-2020-042417 This manuscript describes the results of a study that examined the rate of neuropsychiatric adverse events associated with varenicline use in the general population and among those with COPD (and those with and without psychiatric illnesses). While there is growing and relatively strong consensus on the safety of varenicline, additional data from real-world studies such as the one here is useful. Further, the study is strengthened by its inclusion of COPD patients and those with and without a psychiatric history. The sample is very large and the analytic approach is appropriate. Addressing the relatively minor concerns below may strengthen this important study.  1. The statement in the abstract that the data on the neuropsychiatric safety of varenicline is inconsistent is over-stated. The preponderance of the evidence now supports the drug's safety. Consider stating that simply more data from the real-world can strengthen knowledge. 2. It was not clear to me that the comparisons were largely based on contrasting events with NRT. This should be indicated in the manuscript title or at least made clearer in the abstract. 3. In the Introduction, the statement that evidence from real world studies is lacking is inaccurate. There are several observational studies using real world data on this topic (e.g., Kaduri et al., 2015; Wang et al., 2013). 4. I am not familiar with the term inception to describe a cohort study. Consider removing or defining. 5. There is no certainty that the participants actually took the medications (varenicline or NRT)? Emphasize as potential limitation. 6. Postal code as a measure of SES seems weak. Comment, justify or consider as limitation.
---

REVIEWER	Wenbin Liang Curtin University, Australia
REVIEW RETURNED	29-Sep-2020

GENERAL COMMENTS	Interesting paper, application of treatment-effects estimation for observational data is recommended for the statistical analysis of this study. For example, it would be appropriate to apply inverse-probability weighting, propensity-score matching or inverse-probability-weighted regression adjustment in the analysis. This is because that the choice of treatment (i.e. varenicline or NRT) was likely based on mental health status of the patient when the treatment plan was determined. As it is showed in Table 1, among the “general population”(patients without COPD treatments): $1427/(1427+4480) = 24.16\%$ of the varenicline group patients have psychiatric conditions at baseline (before “index date”) vs $1200/(1200+1970) = 37.85\%$ of the NRT group patients. This indicates that patients prescribed varenicline were less likely to receive drug treatment for psychiatric conditions at baseline. In this study, outcome of interest, psychiatric conditions at “baseline” and COPD status were determined using medication prescription records instead of clinical diagnoses.
--

VERSION 1 – AUTHOR RESPONSE

Reviewer #1:

Reviewer Name: A. Eden Evins

This is a report of an apparently well-executed investigation of subsequent psychiatric medication use within 12 and 24 weeks of initiating varenicline or NRT for smoking cessation, looking particularly at smokers with COPD. It is generally clearly written. I have a number of small comments below. The primary comment I have is to improve the precision with which the outcome variable is described. The outcome is a new psychotropic medication, and should be described as such throughout. A table is needed to describe the actual outcomes (new medications) and if further inference re NPSAE is made how each new medication was interpreted (what AE each medication is inferred to be associated with).

Response: We appreciate this comment, and we agree that improving the precision of the description of outcome variable is necessary. We have adjusted our manuscript further to make the outcome variable clearer for readers by (1) adding the medication name under the ATC codes level that we used in supplementary table S1 to present how these outcome events were exactly defined. (2) adding additional notes under the main Table 2 and Table 3 to remind readers again about how our NPAEs were defined. (3) adding the medication name under the ATC code level that we used, along with related ATC codes in “Exposure and outcomes”. (4) adding additional information in “Exposure and outcomes” to further explain definitions of NPAE “Those who used the drugs under the above-mentioned ATC codes level for NPAEs that prescribed before index date but without new prescription after index date were not count as people with NPAE. Those who never be prescribed drugs under above-mentioned ATC codes level before the index date, but were prescribed (naive prescription) after index date will be included as people with NPAE.”

Of note, different from our definition for psychiatric/non-psychiatric cohorts using general ATC codes, we only used specific ATC codes to define the NPAE including depression, anxiety, and insomnia only. So, our outcome could not be easily described as “a new prescription of psychotropic medication”, the better expression should be “a new or naive prescription of specific medications for depression, anxiety or insomnia within 24 weeks of index date”. But it is long and not suitable for the variable name of outcome. That is why we made the above four adjustments, rather than directly

changing the name of outcome variable as you suggested. The more detailed considerations and explanations about this point could be found in the related response below.

Abstract

Objectives: Data on neuropsychiatric safety of varenicline is remarkably consistent outside of single case reports, particularly for high-risk populations. The first sentence should be deleted. If there is a reason to believe NPSAEs would be different in those with and without COPD, by psychiatric diagnosis, it should be clarified here.

Response: As suggested, we have deleted the first sentence and modified the Objective as "To evaluate the real-world association between varenicline and neuropsychiatric adverse events (NPAEs) in general and COPD population with and without psychiatric disorders compared with nicotine replacement therapy (NRT) to strengthen the knowledge of varenicline safety."

Design: Please identify the comparison groups here, defining the psychiatric subcohort as people in the dataset on two or more psychotropic medications.

Response: As suggested, (1) we have indicated the NRT as the comparison group in the Objective part of Abstract by stating that "To evaluatecompared with nicotine replacement therapy (NRT) to strengthen knowledge of varenicline safety." (2) Definition of psychiatric sub-cohort was added in Participants part of Abstract by stating that ".....Psychiatric sub-cohort was defined as people prescribed psychotropic medications (≥ 2) within 6 months before the index date."

Participants: Please clarify here: Are participants all adults in the dataset who are new users of varenicline or NRT with and without COPD?

Response: yes, all participants are adults in the datasets who are new users of varenicline or NRT. Of note, COPD patients were adults of higher age. To make it clear, this part has been revised as "New users of varenicline or NRT among general (≥ 18 years) and COPD (≥ 40 years) population"

Outcome measures: please state the source of the NPSAE data here and begin to define it as a new psychotropic medication.

Response: Thank you for this point, (1) The source of NPAE data is the IADB.nl prescription database, which had been introduced in the Setting part of Abstract that "Prescription database IADB.nl, Netherlands" (2) we also have revised this part by mentioning data source IADB.nl further and added related definitions that "Outcome measures: The incidence of NPAEs including depression, anxiety, and insomnia, defined by new or naive prescriptions of related medications in IADB.nl within 24 weeks after first treatment initiation of varenicline or NRT."

Results: That varenicline was associated with a significantly lower risk for a new psychiatric medication than the referent NRT group in several general population subsamples should be noted.

Response: We agree that there is a lower risk for at least one subgroup NPAEs (anxiety, depression, or insomnia) by varenicline compared with NRTs in all these four cohorts. Thus, we noted this information in the Result part by adding that "Results from subgroup or sensitivity analyses also did not reveal increased risks of NPAEs, but showed decreased risk of some subgroup NPAEs associated with varenicline."

Conclusion: A stronger statement such as this one from the conclusion would convey the findings better: In contrast to the concerns about a possible increased risk of NPAEs among varenicline users,

we found a 19% and 22% relative decrease in NPAEs in varenicline users of the general population with and without psychiatric disorders, respectively, compared with NRT and no difference in the smaller sample with COPD.

Response: As suggested, we have adjusted our conclusion with a stronger statement that “In contrast to the concerns of a possible increased risk of NPAEs among varenicline users, we found a relative decreased risk of total NPAEs in varenicline users of the general population in psychiatric or non-psychiatric cohorts compared with NRT and no difference for NPAEs between varenicline and NRT users in smaller population with COPD.”

Introduction

The review of the literature has errors and should be reviewed. I believe varenicline was not the first non-nicotine pharmacotherapy for smoking cessation. The early safety concerns regarding varenicline were from uncontrolled case reports and should be so stated.

Response: Thank you for pointing out the errors. (1) "first" is a typing mistake, it should be "first-line", we have corrected it in revised version. (2) we have restated the point you mentioned by changing previous expression into "Of note, the early safety concerns regarding varenicline were from uncontrolled case reports, which could not establish the causality because of a lack of control or comparator."

Re: related evidence from real-world setting is still lacking.

The several high quality large observational trials comparing NPSAEs in varenicline and NRT treated smokers, all of which reported finding no difference, should be cited, eg Gunnell 2009, Meyer et al 2012, the US VA and DOD observational studies.

Response: We agree with you that our previous expression is improper. Here what we would like to express is that “related evidence about varenicline neuropsychiatric safety in high-risk population (e.g., those with COPD or psychiatric disorders or both) from a real-world setting is only explored in few studies”. As you could see that the whole paragraph is discussing the real-world evidence of varenicline safety in high-risk population.

To avoid further misunderstanding, we made the following revisions.

(1) We replace the previous expression by “real-world varenicline safety among these high-risk population is only explored in few studies,28,29 several related large observational studies of high-quality were all about general population, rather than specific high-risk population.30-33”

(2) We inserted the four important references you mentioned.

Of note, these four studies you mentioned are all about general population, although “US VA Cunningham 2016”and “Meyer 2012” mentioned that their included patients are those with and without psychiatric comorbidities, they did not sperate the whole population into two cohorts based on their psychiatric disorder status (psychiatric cohort, non-psychiatric cohort), The result is still about the whole general population, rather than the specific subgroups with high-risk population (those with psychiatric disorder only).

Re: Only few studies assessed the neuropsychiatric safety of varenicline in patients with COPD or psychiatric disorders,10,20-22 and Re: results were inconsistent;

Analyses of EAGLES trial NPSAEs within psychiatric subgroups vs non psychiatric controls, eg. Heffner et al 2019 and Evins et al 2020 should also be cited. These report clinically ascertained NPSAEs at a rate much lower than reported in this paper. The Chengappa et al 2014 J Clin Psych paper of a varenicline trial in 60 people with bipolar disorder should also be cited. Trials of varenicline in schizophrenia and or bipolar disorder should also be cited: Evins et al JAMA 2014, Williams et al J Clin Psych, Pachas et al 2012 J Dual Diag, and in MDD: Antheneli Ann Int Med 2013.

The NPSAE safety results from the above trials and large observational studies with NRT reference groups are remarkably consistent. The overwhelming bulk of controlled trial and observational data shows NPSAE risk of varenicline to be no different from or less than NRT and/or placebo. This is reviewed in Cather et al 2017 CNS Drugs for those with psychiatric illness and should be so stated. This does not diminish the importance of an investigation of psychiatric safety of varenicline in a COPD sample with and without evidence for psychiatric comorbidity. Please also mention the EMA removal of the boxed warning in addition to the FDA action.

Response: Thank you for these comments with useful references. Like my response to your previous comment, here we would like to say “only few studies from real-world setting assessed the neuropsychiatric safety of varenicline in patients with COPD or psychiatric disorders.....” As you could see that the whole paragraph is discussing the real-world evidence of varenicline safety in high-risk population. Based on your suggestion, we made the following revision.

(1) We replaced the previous expression with that “Although many RCTs and post hoc analyses of some RCTs showed consistent results about varenicline safety in subjects with specific psychiatric disorder,20-27 real-world varenicline safety among these high-risk population is only explored in few studies,28,29 several large observational studies of high-quality were all about general population, rather than specific high-risk population.30-33 It is important to further investigate the psychiatric safety of varenicline in special patients with COPD or psychiatric disorder or both in real-world setting to strengthen the knowledge of varenicline safety.”

(2) We inserted these clinical trials and high-quality observational studies you mentioned in the above contents.

(3) We have deleted the improper expression that “the results were inconsistent”

(4) We added the removal of varenicline by EMA, along with action of FDA that “Although the Black Box warning in varenicline’s product labelling was already removed by both FDA and European Medicines Agency (EMA) in 2016, there are still concerns.....”

Re: we classified the individuals into a psychiatric cohort and non-psychiatric cohort according to the presence of psychiatric disorders defined by the prescription of two or more drugs from the neurological ATC group, i.e., N02, N03, N04, N05, N06 within 6 months before index date. Please list these medications.

Response: As suggested, (1) we listed the medication name of the 3rd ATC level, consistent with the ATC codes we used, in the revised manuscript that “.....defined by the prescription of two or more drugs of analgesics (ACT codes N02, see supplementary Table S1), antiepileptics (N03), anti-parkinson drugs (N04), psycholeptics (N05) or psychoanaleptics (N06)”; (2) As the specific medication name lists of the final/7th level (e.g. N02AA01) under the 3rd level ATC code (e.g. N02) are very long, so we did not present these very exact medication name lists in the main text. (2) However, we also listed the medication name of 3rd ATC level, consistent with the ATC codes we used, in supplementary Table S1 and attached the link (https://www.whooc.no/atc_ddd_index/) of WHO ATC system as a note under the Table S1 for readers to easily check the very specific medication name lists of lower level under ATC codes we listed if they are interested.

A new psychiatric medication within 6 months of starting NRT or varenicline among those already on 2 or more psychiatric medications is a crude measure for a neuropsychiatric adverse event. This should be discussed in the limitations section. Psychiatric medication changes are common and may be due to many reasons in addition to worsening of psychiatric symptoms, for example common reasons for changing psychiatric medications are sexual or anticholinergic side effects, sedation, extrapyramidal symptoms etc. Please refer to this outcome in the paper not as a NPSAE but as a new psychiatric medication. Thus, depression and insomnia events would be new antidepressant medication or sedative-hypnotic medication.

Response: As suggested, (1) We have mentioned the limitation by stating that “as outcome events ...were defined by prescriptions of related drugs as proxies, which is a crude measure and may have led to some misclassification” (2) For problem about psychiatric medication change, this could happen in patients in psychiatric cohort (with previous psychiatric prescriptions), we also considered this problem before. That is why we set a sensitivity analysis “to exclude an active psychiatric status by limiting the population to those who were not prescribed any drugs for psychiatric disorders within 30 days before index date”, which means the new prescriptions for our defined outcome events have very high chance to be influenced by Varenicline or NRTs, rather than psychiatric changes. These sensitivity analysis in general and COPD patients were presented in first situation in Table S4 and Table S6 of supplement file, respectively. They are rather consistent with our result showed in Table 2 and Table 3 of main text; (3) Another consideration is that psychiatric medication change seems not be a problem at all for those in non-psychiatric cohort, as they were not been previously prescribed or only been prescribed once psychiatric medication according to our definition, therefore, the new prescription or naive prescription has very low chance to be influenced by psychiatric medication change.

For your suggestion to change name of NPAEs, besides our explanation mentioned in (2) and (3), we also have the following three considerations:

(i) In Abstract, “Exposure and Outcomes” part of Methods, and limitation part of Discussion, we all have stated clearly that NPAEs including depression, anxiety and insomnia were defined by new or naive prescription of related drugs within 24 weeks after index date of first treatment initiation of varenicline and NRT use;

(ii) For overall NPAEs, we could not use “a new psychiatric medication” to describe it simply, first, as we notice that for those in non-psychiatric cohort that did not have previous psychiatric medication use, the prescription within our observation periods should be “first or naïve”, rather than “new”; we have added “new or naïve” throughout in the revision for our definition and explain it further. secondly, psychiatric medication is a very general word, which we use it define our cohorts (psychiatric or non-psychiatric cohort), but we did not use the general ATC codes for our NPAE outcome, so “psychiatric medication” seems not accurate, too; If we have to define it by using medication, I think “a new or naïve related psychiatric medication prescription for depression, anxiety, insomnia” may be proper, but this expression is long and will distract readers attention.

(iii) for subgroup outcome events like depression, we defined it not only use antidepressants (ATC code N06A) itself, but also included antidepressants in combination with psycholeptics (ACT code: N06CA), so if we name the depression event with medications, it will be “a new or naïve prescription of antidepressants or antidepressants in combination with psycholeptics”, it is also very long.

(iv) The use of these medications to replace the NPAE outcome name is also inconsistent with expression of previous studies and make it difficult for readers to understand.

Therefore, based on above consideration, we decided not to change the current expression of these events directly, but to revise our manuscript from the following aspects to make our outcome definition clearer for readers:

a) In “exposure and outcomes” part of Methods, we added these medication names of ATC 4th or 5th level (like you suggested before) used for definition of our three focused events: depression, anxiety and insomnia.

b) we also give the exact medication name of the ATC codes level we used, along with related ACT codes to explain our definitions of NPAEs in Table S1 of supplementary file.

c) we have added the “naive” along with “new” for definition NPAE that “one or more new or naive prescription of.....” throughout the manuscript about NPAE.

d) we also further explain that “Those who used the drugs under above-mentioned ATC codes level for NPAEs that prescribed before index date but without new prescription after index date were not count as people with NPAE, those who never be prescribed drugs under above-mentioned ATC codes level before index date, but be prescribed (naive prescription) after index date will be counted as people with NPAE.”

Also, it is not clear why pain medications N02 are included, N03 anti-epileptic drugs would indicate the need for a mood stabilizer and would be associated more with bipolar disorder than depression, anxiety, or insomnia. N05 includes antipsychotics and would be associated more with bipolar disorder or schizophrenia than depression, anxiety, or insomnia.

Response: To make sure there is no misunderstanding here. We would clarify that N02-N05 were only used for the definition of psychiatric disorders as baseline for us to classify the population into psychiatric (high-risk) and non-psychiatric cohorts, rather than definition for our three NPAE outcomes events (depression, anxiety, insomnia) directly. Based on this understanding, we have several considerations to include N02, N03, and N05 to define our psychiatric and non-psychiatric cohorts (1) pain medication N02 is just as proxy for some chronic pain like Migraine, the migraine patients often suffer from symptoms of depression and anxiety, which are worsen when migraine attacks occur very often. (2) like what you said, N03 and N05 are more associated with bipolar disorder, which is a disease with mood change up and downs, which may also influence the outcome we focused. (3) To keep consistent with previous criteria that used as identification of psychiatric cohort, we also used a broad way to include all possible mental problems as definition for the high-risk population for NPAEs, as those with any mental disorder were underutilized of varenicline, that is also why we need to explore the evidence among all these special population.

In defining the sample, those on two or more psychiatric medications are not known to have a psychiatric diagnosis. They can be referred to as those on psychiatric treatment. The difficulty here is that for example SSRIs are listed by the WHO as a treatment for depression but are often used to treat anxiety. Trazodone is an antidepressant may be used more commonly for insomnia than depression. Thus, for defining the sample and the outcome, using the medications rather than the illness would improve the precision of the report.

Response: we totally understand your suggestion. In fact, we also considered this before. But we give up this naming based on the following reasons. (1) First, for the practicability using medication name, rather than illness for outcome events, we have explained before in previous response for reasons of our decision, and have taken some revision to make it clear. (2) for the possibility of using medication treatment, rather than illness to define the sample of population (psychiatric, non-psychiatric), like you suggested "those on psychiatric treatment", we also think it is not accurate in our case. As you could see we use two or more psychiatric medications (based on references) to define the psychiatric disorder at baseline, so those with only one medication prescription were treated as non-psychiatric cohort as the chance of psychiatric diagnosis is very low. Thus, if we name the sample/subjects in psychiatric cohort as those on psychiatric treatment, it is not consistent with our above definition, as there is possibility that patients in non-psychiatric cohort may also use psychiatric treatment, although could be just once. (3) The way we used the two or more medications for identifying psychiatric high-risk is based on published references, as two or more prescriptions of related medication will have much higher chance to have diagnosis psychiatric disorder than only once, number of medication prescriptions also present the severity of psychiatric disorders. We have added the possibility of misclassification as one of limitations in the discussion.

Results

Those on psychiatric medication may be less likely to receive varenicline than those not on psychiatric treatment. Please report whether there is an interaction between psychiatric medication vs not and prescription of varenicline vs NRT.

Response: As suggested, we have rechecked our analyses and added the interaction between treatments (varenicline vs NRT) and psychiatric medication (psychiatric vs non-psychiatric) in the model among the general and COPD population, respectively. we did not find their significant

interaction effects. (1) we reported this result “In the model for the primary outcome of NPAEs among overall general population, there were no significant interaction between treatments (varenicline vs NRTs) and cohorts (psychiatric vs non-psychiatric) interaction, which is also the same for the model of NPAEs among overall COPD patients.” in our result part. (2) like you concerned that those on psychiatric medication may be less likely to receive varenicline than those not on psychiatric treatment, to reduce this possible selection bias from general practice, we also conducted a regression adjusted by inverse probability of treatment weighting (IPTW), which is suggested by another reviewer, as sensitivity analyses of our main results. The results from IPTW is consistent with our main result and has been added in supplementary document Table S4 and S6 in revised version.

Discussion

Re: Of note, our result is consistent with another cohort study based data from the general practice that no increased depression were observed to be associated with varenicline (HR 0.88 [0.77-1.00]).
33

The findings are consistent also with those of Cather et al J Dual Diag 2017, which should be cited, finding varenicline treatment associated with reduced depressive symptoms. Because of higher rates of abstinence, this is also consistent with a statement later in the manuscript: Moreover, there is evidence that quitting smoking is associated with recovery in stress, anxiety and depression in smokers.^{45,46}

Response: Thank you for sharing this reference, which is relevant to these two discussion points. we have cited this paper by Cather et al 2017 in these two places as suggested.

Re: However, the rate of NPAEs was substantially higher in participants with psychiatric disorders than those without such illness, which was also consistent with findings from previous studies.³⁷⁻³⁹ This is not consistent with trials of varenicline in people with actual diagnosed psychiatric illness as cited above in a dozen studies in trials and observational data in the US and UK. Likely this is related to the definition of a medication change as a ‘NPSAE’ which could both be unrelated to varenicline or NRT and could often miss NPSAEs related to varenicline or NRT.

Response: To make sure there is no misunderstanding here, we would like to stress that we did not mean NPAEs between varenicline vs NRT here, but to compare the NPAEs rates between psychiatric cohort (those with psychiatric disorder) and non-psychiatric cohort (those without psychiatric disorder).

(1) Although lots of clinical trial and several observational studies were conducted among those with psychiatric disorder, only few studies presented and compared the NPAEs outcomes in both psychiatric and non-psychiatric cohorts. [1-4]

(2) We found that NPAEs rates were higher in psychiatric cohort than non-psychiatric cohort, this was definitely consistent with ENGLES RCT by Anthenelli et al in 2016 , which stated in results part that “those in the psychiatric cohort were more likely to report neuropsychiatric adverse events of all types than those in the non-psychiatric cohort”; [1] same results and expression were also found in three post-hoc/subgroup analyses of ENGLES with similar statements such as “There was a main effectwith a higher overall incidence of NPSAEs in the BD (bipolar disorder) sub-cohort versus the NPC (non-psychiatric cohort)”[2-4]; The rates difference between two cohorts were also could easily be found in their related tables.

(3) The several observational studies you mentioned before like Gunnell, et al BMJ 2009 (UK, GPRD), Cunningham, et al. Addiction 2016 (US, VA), [5,6] they did not present and compare the results in sub-cohorts (those with psychiatric disorder, those without psychiatric disorder). Thus, we could not compare our outcomes comparisons between sub-cohorts with the results from these observational studies.

(4) we admit that we could not totally exclude the possibility that you mentioned about medication change. However, to exclude an active psychiatric status with possible frequently medication change, we have done the sensitivity analyses by further limiting the study population only to those subjects who were not prescribed any drugs for psychiatric disorders within 30 days before index date. The sensitivity results were still similar and consistent with our main results showed in table 2 and d3, as well as consistent with results of ENGLES. At least this means the chance like what you said is very small.

Based on the above consideration, we made the following revisions: (i) we revised the related expression that “however, the rates of NPAEs were substantially higher among participants in psychiatric cohort than those in non-psychiatric cohort, which were also consistent with findings from previous studies.” (ii) we have cited following references [1-4] in revised version.

References:

[1] Anthenelli RM, Benowitz NL, West R, et al. Neuropsychiatric safety and efficacy of varenicline, bupropion, and nicotine patch in smokers with and without psychiatric disorders (EAGLES): a double-blind, randomised, placebo-controlled clinical trial. *Lancet*. 2016;387(10037):2507-2520.

[2] Heffner JL, Evins AE, Russ C, et al. Safety and efficacy of first-line smoking cessation pharmacotherapies in bipolar disorders: Subgroup analysis of a randomized clinical trial. *J Affect Disord*. 2019;256:267-277.

[3] Ayers CR, Heffner JL, Russ C, et al. Efficacy and safety of pharmacotherapies for smoking cessation in anxiety disorders: Subgroup analysis of the randomized, active- and placebo-controlled EAGLES trial. *Depress Anxiety*. 2020;37(3):247-260.

[4] Evins AE, West R, Benowitz NL, et al. Efficacy and Safety of Pharmacotherapeutic Smoking Cessation Aids in Schizophrenia Spectrum Disorders: Subgroup Analysis of EAGLES. *Psychiatr Serv*. 2020:appips202000032.

[5] Gunnell D, Irvine D, Wise L, Davies C, Martin RM. Varenicline and suicidal behaviour: a cohort study based on data from the General Practice Research Database. *BMJ*. 2009;339:b3805.

[6] Cunningham FE, Hur K, Dong D, et al. A comparison of neuropsychiatric adverse events during early treatment with varenicline or a nicotine patch. *Addiction*. 2016;111(7):1283-1292.

Conclusion:

It should be noted that not only did varenicline cause no greater new psychotropic medication use in your COPD sample of interest, it was associated with statistically significantly lower incidence of new psychotropic drug use in several subgroups. In particular, in the non-COPD psychiatric cohort, varenicline was associated with significantly fewer new psych meds than NRT. This is notable and consistent with the prior literature.

Response: as suggested, we have added the statement “Of note, varenicline were even associated with lower risk of some subgroup NPAEs than NRTs in sub-cohorts of our population, our findings are notably and consistent with previous literature” In conclusion part.

Table 2: please list categories and individual drugs instead of disorders. Many medications can be used to treat two or more of these disorders. Serotonin and serotonin norepinephrine reuptake inhibitors may be used to treat anxiety and or depression. Sedative hypnotics may be used to treat anxiety or insomnia. Was there no new antipsychotic or mood stabilizing treatment noted?

Response: This is a question that raised before, we have given our responses. As it is difficult to use medications to define these mental disorders exactly, thus, we used the related ATC codes that consistent with previous published literatures to define Depression (ATC codes: N06A, N06CA), anxiety (N05B), and insomnia (N05C), although it is unavoidable to result in some misclassification, which we have mentioned as one of our limitation. (1) for your question, In ATC system, serotonin reuptake inhibitors belong to antidepressants (N06A), that is why we use it to define depression only,

rather than anxiety. (2) hypnotics and sedative (N05C) were used to define insomnia only in this study, rather than anxiety. (3) Psycholeptic has been used to define depression in this study, as it is clear that “antidepressants in combination with psycholeptic” were classified under N06CA in ATC classification system.

However, we also consider these possible misclassifications, (i) that is why we also have our overall NPAEs outcome, which is defined as new or naïve related medication prescription for depression, anxiety or insomnia within 24 weeks after index date. NPAEs overall outcome (first line result in Table 2) could capture all related medication for our three focused events. (ii) like you suggested, we have added note in both table 2 and table 3 to remind readers again how we define NPAEs by prescription of related medications. The medication category and related ATC codes for NPAE outcomes have already been introduced in our Methods part. (iii) The individual drug lists under the ATC codes level we used to define our NPAEs were long (and seems not appropriate to present in Table 2 and Table 3, thus, we presented the medication category in supplementary table and give notes that the detailed medication lists under exact ATC code level could be easily found in WHO website https://www.whooc.no/atc_ddd_index/ ; (e.g. put in ATC code N06A, there are five sub-levels, under each fifth level, 10-25 medications were listed, that is why in study like us based on prescription database, we usually stated clearly what ATC codes we used, rather than the exact medication names)

Reviewer #2

Reviewer Name: Robert Schnoll

This manuscript describes the results of a study that examined the rate of neuropsychiatric adverse events associated with varenicline use in the general population and among those with COPD (and those with and without psychiatric illnesses). While there is growing and relatively strong consensus on the safety of varenicline, additional data from real-world studies such as the one here is useful. Further, the study is strengthened by its inclusion of COPD patients and those with and without a psychiatric history. The sample is very large and the analytic approach is appropriate. Addressing the relatively minor concerns below may strengthen this important study.

1. The statement in the abstract that the data on the neuropsychiatric safety of varenicline is inconsistent is over-stated. The preponderance of the evidence now supports the drug's safety. Consider stating that simply more data from the real-world can strengthen knowledge

Response: Thank you for this point, we have deleted the previous statement and revised this part as you suggested that "Objective: This study aims to evaluate the real-world association..... to strengthen the knowledge of varenicline safety."

2. It was not clear to me that the comparisons were largely based on contrasting events with NRT. This should be indicated in the manuscript title or at least made clearer in the abstract.

Response: As suggested, we have added the comparison information in abstract that "This study aims to evaluatecompared with nicotine replacement therapy (NRT)" to make it clear to readers.

3. In the Introduction, the statement that evidence from real world studies is lacking is inaccurate. There are several observational studies using real world data on this topic (e.g., Kaduri et al., 2015; Wang et al., 2013).

Response: This expression is improper and was also raised by another reviewer. Here we would like to say that “related evidence about varenicline neuropsychiatric safety in high-risk population (e.g., those with COPD or psychiatric disorders or both) from real-world setting is only explored in few studies”. As the whole paragraph is discussing the real-world evidence of varenicline safety among

these high-risk population in real-world setting. (1) lots of real-world observational studies (like references what you mentioned) were conducted among general population, rather than high-risk subgroup populations like those with psychiatric disorder or those with COPD or both, although lots of clinical trials have focused these population. (2) To avoid misunderstanding, we replaced the previous expression by "Although many RCTs and post hoc analyses of some RCTs showed consistent results about varenicline safety in subjects with specific psychiatric disorder,20-27 real-world varenicline safety among these high-risk population is only explored in few studies,28,29 several related large observational studies of high-quality were all about general population, rather than specific high-risk population.30-33 It is important to further investigate the psychiatric safety of varenicline in special patients with COPD or psychiatric disorder or both in real-world setting to strengthen the knowledge of varenicline safety." (3) we also inserted related literatures based on suggestions from you and previous reviewer.

4. I am not familiar with the term inception to describe a cohort study. Consider removing or defining.

Response: we would like to use "inception cohort" to express the start point of this cohort was onset (first time) of the smoking cessation and start (first time) of therapy with varenicline or NRT. Consider it is not common term to use. We have removed it throughout this revised manuscript to avoid possible misunderstanding.

5. There is no certainty that the participants actually took the medications (varenicline or NRT)? Emphasize as potential limitation.

Response: In the "Strength and limitations" part, we have added this as one of limitations by stating that "Besides, the prescription of medication may not always lead to actually intake of the drugs (non-adherence) and such misclassification....."

6. Postal code as a measure of SES seems weak. Comment, justify or consider as limitation.

Response: we agree with you that postal code as a proxy for SES could not totally reflect the real SES and may led to some misclassification. We added it as limitation in the revised manuscript by stating that " Lastly, SES was determined by the postal code level, which could lead to some misclassification, although postal code was a frequently used predictor of SES.60"

Reviewer: 3

Reviewer Name: Wenbin Liang

Interesting paper, application of treatment-effects estimation for observational data is recommended for the statistical analysis of this study. For example, it would be appropriate to apply inverse-probability weighting, propensity-score matching or inverse-probability-weighted regression adjustment in the analysis. This is because that the choice of treatment (i.e. varenicline or NRT) was likely based on mental health status of the patient when the treatment plan was determined. As it is showed in Table 1, among the "general population" (patients without COPD treatments): $1427/(1427+4480) = 24.16\%$ of the varenicline group patients have psychiatric conditions at baseline (before "index date") vs $1200/(1200+1970) = 37.85\%$ of the NRT group patients. This indicates that patients prescribed varenicline were less likely to receive drug treatment for psychiatric conditions at baseline.

Response: we admit that selection bias may existed in this real-world study, although the remove of black-box warning, GP may still be reluctant to prescribe varenicline to those with psychiatric disorders, which is also reflected in our baseline characteristics. Our baseline characteristics are also consistent with previous studies.

As suggested, we have re-run our analyses in our main results presented in table 2 and table 3 by applying the inverse-probability weighted (IPW) regression. (1) The results by IPW are consistent with results in table 2 and 3 and the conclusion did not change. Of note, for the general population with psychiatric disorder, results by IPTW adjustment even further support the benefits of mood change by varenicline and showed that the depression and insomnia were also reduced with boundary significance by varenicline compared with NRTs. (2) We have added IPW regression as sensitivity analyses in “methods” Part, and put the results by IPW regression in the supplement document in Table S4 and Table S6 as extra situation. (3) we also added the results by IPTW regression in “Sensitivity results” part by mentioning that “After adjusting the results by IPTW regression in cohorts of general populations, the results are consistent with the results presented above. Also, no significant change was observed by IPTW regression for cohorts of COPD patients compared with results above.”

In this study, outcome of interest, psychiatric conditions at “baseline” and COPD status were determined using medication prescription records instead of clinical diagnoses.

Response: Like all studies based on prescription database only, the database did not have information of clinical diagnoses. This is a limitation which we have added in "Strength and Limitations" part.